# Sequence variation and immunogenicity of the *Mycoplasma genitalium* MgpB and MgpC adherence proteins during persistent infection of men with non-gonococcal urethritis

Gwendolyn E. Wood[1]*, Stefanie L. Iverson-Cabral[1], Catherine W. Gillespie[2], M. Sylvan Lowens[3], Lisa E. Manhart[4,5], Patricia A. Totten[1,6]

1 Department of Medicine, Division of Allergy and Infectious Diseases, University of Washington, Seattle, Washington, United States of America, 2 Institute for Health Metrics and Evaluation, University of Washington, Seattle, Washington, United States of America, 3 Public Health - Seattle & King County Sexual Health Clinic, Seattle, Washington, United States of America, 4 Department of Epidemiology, University of Washington, Seattle, Washington, United States of America, 5 Department of Global Health, University of Washington, Seattle, Washington, United States of America, 6 Department of Global Health, Pathobiology Interdisciplinary Program, University of Washington, Seattle, Washington, United States of America

* gwenwood@uw.edu

## Abstract

*Mycoplasma genitalium* is a sexually transmitted bacterial pathogen that infects men and women. Antigenic variation of MgpB and MgpC, the immunodominant adherence proteins of *M. genitalium*, is thought to contribute to immune evasion and chronic infection. We investigated the evolution of *mgpB* and *mgpC* sequences in men with non-gonococcal urethritis persistently infected with *M. genitalium*, including two men with anti-*M. genitalium* antibodies at enrollment and two that developed antibodies during follow-up. Each of the four patients was persistently infected with a different strain type and each patient produced antibodies targeting MgpB and MgpC. Amino acid sequence evolution in the variable regions of MgpB and MgpC occurred in all four patients with changes observed in single and multiple variable regions over time. Using the available crystal structure of MgpC of the G37 type strain we found that predicted conformational B cell epitopes localize predominantly to the variable region of MgpC, amino acids that changed during patient infection lie in these epitopes, and variant amino acids are in close proximity to the conserved sialic acid binding pocket. These findings support the hypothesis that sequence variation functions to avoid specific antibodies thereby contributing to persistence in the genital tract.

## Introduction

*M. genitalium* is increasingly recognized as a sexually transmitted pathogen in men as a frequent cause of acute and chronic non-chlamydial, non-gonococcal urethritis (NGU) [1]. In

**Data Availability Statement:** All relevant data are within the manuscript and its Supporting Information files.

**Funding:** Funding provided by the National Institute of Health, NIAID grants R21 AI107402 (GEW, SLIC, PAT), U19 AI031448 (SLIC, LEM, MSL, CWG), and R01 AI072728 (LEM, PAT). The funders had no role in study design, data collection and analysis, decision to publish, or preparation of the manuscript.

**Competing interests:** The authors have declared that no competing interests exist.

women, *M. genitalium* is associated with cervicitis, pelvic inflammatory disease (PID), endometritis [2–4], tubal factor infertility [5, 6], preterm birth, and spontaneous abortion [7]. Importantly, *M. genitalium* infection increases cervical shedding of HIV [8] as well as the risk of acquiring and transmitting HIV [9, 10]. The prevalence of *M. genitalium* ranges from 1.3–3.9% in population-based studies, to 20.5% in high risk settings [11, 12]. Similar to *Chlamydia trachomatis* [13], many infected men and women are unaware of their *M. genitalium*-positive status [12, 14, 15]. Treatment is complicated by inherent resistance to cell wall targeting antibiotics, and high rates of acquired azithromycin resistance (40–100% of strains in some settings) [16]. The efficacy of moxifloxacin, used to treat azithromycin-resistant infections, is declining and dual resistance is increasingly reported [17].

*M. genitalium* can persist for months, and potentially years, in infected individuals [10, 18, 19] despite the presence of specific antibodies in genital exudates of infected women [20] and in the sera of infected men [21]. These data suggest that *M. genitalium* evades the local and systemic immune response, allowing greater opportunity for transmission to others and ascension to the upper reproductive tract in women. The MgpB and MgpC adherence proteins, also known as P140/MG191 and P110/MG192, respectively, are immunodominant targets of host antibodies [5, 20–23]. MgpB and MgpC localize to the *M. genitalium* terminal organelle, forming a complex [24] required for adherence to host cells and inanimate surfaces, and motility [25–28]. Recently the structure of the MgpC protein was determined and a sialic acid binding pocket was identified [29].

The *mgpB* and *mgpC* genes, expressed from a single locus on the *M. genitalium* chromosome, consist of conserved sequences interspersed with variable regions [22, 23, 30]. MgpB and MgpC expression is affected by both antigenic and phase variation. Antigenic variants express MgpB and MgpC proteins with variant amino acids while phase variants do not express either MgpB or MgpC and are non-adherent. Antigenic variation is accomplished through segmental, reciprocal recombination between individual variable regions in *mgpBC* and archived variable sequences present in nine MgPars located throughout the chromosome [22, 23, 31, 32]. No MgpB or MgpC protein is expressed from the MgPars as only the variable sequences of *mgpB* and *mgpC* are present, the adjacent variable regions have different reading frames, and the variable sequences are often separated by short AT-rich regions encoding multiple stop codons. Phase variants arise *in vitro* by multiple mechanisms including recombination between *mgpBC* and the MgPars, point mutations, and deletions. [25, 33, 34]. Phase variants generated by recombination fall into at least six classes and can be reversible or irreversible depending on the number of recombination partners involved [33]. Deletion of *recA* results in the near-total loss of antigenic and phase variation implicating recombination as the mechanism that generates the majority of these variants [25, 33].

Antigenic and phase variation may represent immune evasion strategies allowing *M. genitalium* to escape binding by specific host antibodies and persist in the genital tract. In order to understand the extent of antigenic variation during infection, we assessed sequence changes in both *mgpB* and *mgpC* in four men with NGU with persistent *M. genitalium* infection [35]. Among these men, two were positive for anti-*M. genitalium* serum antibodies specific for MgpB and MgpC at enrollment and two developed MgpB/C-specific antibodies during observation. We assessed sequence variation simultaneously in all four variable regions: region B, EF, and G of *mgpB*, and region KLM of *mgpC*, and found that variation was both rapid and extensive and was localized to conformational B cell epitopes predicted for MgpC. Our results are consistent with a model in which the immune system selects for variants in multiple regions of MgpB and MgpC simultaneously during persistent infection. Finally, by mapping these sequence changes onto the published crystal structure of MgpC [29], we found that

variant amino acids are located near the sialic acid binding pocket of MgpC, suggesting that antigenic variation may protect *M. genitalium* from adherence-inhibiting antibodies.

## Materials and methods

### Patient specimens

The *M. genitalium* isolates in this study (Table 1) were obtained from urine specimens collected between January 2007 and July 2011 in a double-blinded, randomized trial comparing the effectiveness of azithromycin and doxycycline for men with NGU at the Public Health–Seattle & King County STD Clinic in Seattle, WA [35]. In this study, *M. genitalium* PCR-positive patients were randomly assigned to receive azithromycin or doxycycline upon enrollment (Visit 1). Patients returning at Visit 2 with signs or symptoms of urethritis were prescribed the alternate antibiotic and *M. genitalium* PCR status was again determined. Patients that were *M. genitalium*-positive at Visit 3, after azithromycin and doxycycline treatment, were prescribed moxifloxacin. At each time point, persistent infection was determined by PCR and *M. genitalium* isolates were recovered in cocultures with Vero cells (see below).

Among the four patients whose cultured strains were analyzed in the current study, *M. genitalium* infection persisted after doxycycline treatment, consistent with the known poor efficacy of this antibiotic for eradication of *M. genitalium* infection [36]. Three men (Patients 10378, 10467, and 10477) were also treated with azithromycin per study protocol, however, it was later determined that each of these patients had been infected with an azithromycin resistant strain (MIC $\geq$ 8 µg/ml, Totten *et al*. in preparation) at enrollment. The *M. genitalium* strain that infected Patient 10366 is sensitive to azithromycin, however, azithromycin treatment was initiated after collection of the Visit 3 specimens. For the present study, we analyzed *M. genitalium* isolates cultured from patient specimens obtained at Visit 1 and Visit 3.

### Immunoblots

Antibody reactivity of patient sera to whole cell lysates of wild type *M. genitalium* strain G37 was determined as previously described [38] using a 1:1,000 dilution of patient serum, followed by a 1:7,500 dilution of peroxidase-conjugated goat anti-human IgG (whole molecule; Sigma-Aldrich, St. Louis, MO) secondary antibody and chemiluminescent detection (ECL, GE Healthcare, Chicago, IL).

### Culture of *M. genitalium* from urine

*M. genitalium* isolates (Table 1) were recovered from processed patient urine by coculture with Vero cells [39]. Briefly, patient urine (2 ml) was centrifuged at 16,000 x g for 15 minutes, the supernatant was discarded, and the cell pellet was resuspended in 0.4 ml mycoplasma transport medium and frozen at -80˚C. At the time of culture, 1 x $10^5$ Vero cells (obtained from the American Type Culture Collection) were seeded in 25 $cm^2$ flasks in 5 ml Eagle's Minimal Essential Medium (EMEM; ATCC, Manassas, VA) supplemented with 10% fetal bovine serum and 100 U/ml penicillin. After overnight incubation at 37˚C in 5% $CO_2$, the culture medium was removed, adherent Vero cells were washed with PBS, and fresh EMEM containing 10% FBS, 6% yeast dialysate, 100 U/ml penicillin, 50 µg/ml polymyxin B, and 50 µg/ml colistin in a total volume of 8.5 ml was added. Flasks were inoculated with thawed, processed urine (100 µl) and incubated at 37˚C in 5% $CO_2$ for four weeks. Vero cells grew to form a confluent monolayer after two weeks and then detached from the plastic by week three. To confirm growth of *M. genitalium* from patient specimens, aliquots from these cocultures were collected weekly and DNA was isolated with the MasterPure DNA isolation kit (Lucigen,

**Table 1. M. genitalium clinical isolates used in this study.**

| Patient number | Days between Visit 1 and 3 | Treatments between Visit 1 and 3 | Isolate[a] | Strain type[b] | GenBank Accession Numbers |
|---|---|---|---|---|---|
| 10366 | 33 | Doxycycline | MEGA 1166 | 3 | Region B: MT439353 –MT439366 |
| | | | | | Region EF: MT439373 –MT439393 |
| | | | | | Region G: MT439409 –MT439410 |
| | | | | | Region KLM: MT439412 –MT439443 |
| | | | MEGA 1206 | 3 | Region B: MT439376 –MT439372 |
| | | | | | Region EF: MT439394 –MT439408 |
| | | | | | Region G: MT439411 |
| | | | | | Region KLM: MT439444 –MT439455 |
| 10378 | 48 | Doxycycline | MEGA 1199 | 2 | Region B: MT439456 |
| | | Azithromycin | | | Region EF: MT439458 –MT439459 |
| | | | | | Region G: MT439477 |
| | | | | | Region KLM: MT439479 –MT439484 |
| | | | MEGA 1261 | 2 | Region B: MT439457 |
| | | | | | Region EF: MT439460 –MT439476 |
| | | | | | Region G: MT439478 |
| | | | | | Region KLM: MT439485 –MT439489 |
| 10467 | 28 | Doxycycline | MEGA 1473 | 5 | Region B: MT439490 –MT439502 |
| | | Azithromycin | | | Region EF: MT439518 –MT439528 |
| | | | | | Region G: MT439540 |
| | | | | | Region KLM: MT439542 –MT439544 |
| | | | MEGA 1493 | 5 | Region B: MT439503 –MT439517 |
| | | | | | Region EF: MT439529 –MT439539 |
| | | | | | Region G: MT439541 |
| | | | | | Region KLM: MT439545 –MT439553 |
| 10477 | 50 | Doxycycline | MEGA 1491 | 4 | Region B: MT439554 –MT439555 |
| | | Azithromycin | | | Region EF: MT439560 –MT439571 |
| | | | | | Region G: MT439580 –MT439581 |
| | | | | | Region KLM: MT439583 –MT439591 |
| | | | MEGA 1534 | 4 | Region B: MT439556 –MT439559 |
| | | | | | Region EF: MT439572 –MT439579 |
| | | | | | Region G: MT439582 |
| | | | | | Region KLM: MT439592 –MT439593 |

[a], *M. genitalium* cultured from Visit 1 and Visit 3 urine specimens for each patient.

[b], strain type numbering according to Jensen [37]. Strain type sequences have been deposited in GenBank (KP318822.1, KP318823.1, KP318824.1, KP318825) [27].

Middleton, WI). *M. genitalium* DNA was quantitated by qPCR as previously described [40] confirming growth by an increase in genomes over time.

## PCR amplification and sequencing of *mgpBC* variable and strain typing regions

DNA isolated from *M. genitalium* strains cocultured with Vero cells for three weeks, corresponding to late log phase, served as template for amplification of the *mgpB* strain typing region (Fig 1) with primers ModPetF and 1415R (Table 2). PCR products amplified after 30 cycles with Platinum PCR SuperMix High Fidelity (Invitrogen, Carlsbad, CA) were cloned into pCR2.1-Topo (Invitrogen) and several plasmid clones were sequenced to determine the *M. genitalium* strain type sequence and verify that a single strain type was detected at Visit 1

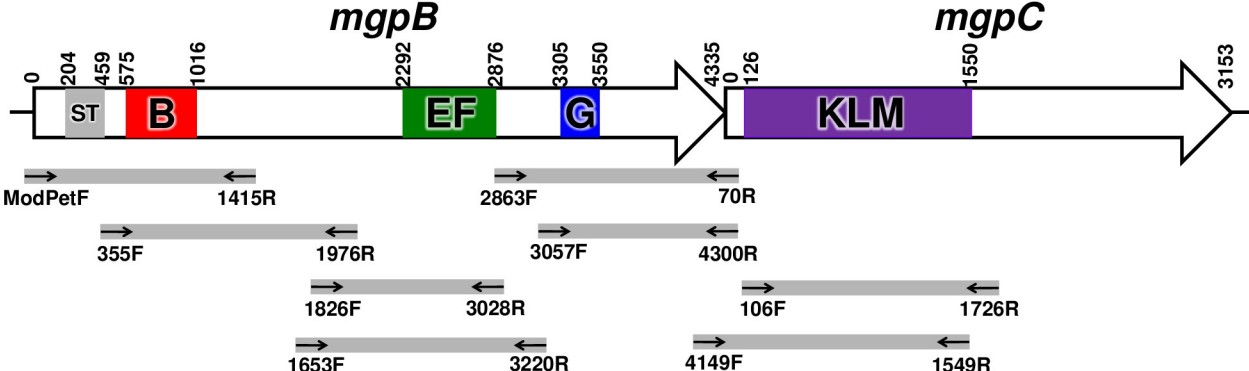

**Fig 1. Strain typing and variable regions of *mgpB* and *mgpC* targeted for PCR amplification and sequencing.** Small arrows indicate the primers used to PCR amplify each region indicated by grey lines. Each variable region was amplified with two different primer pairs. PCR products were cloned and sequenced from individual plasmids to assess sequence changes over time in infected men. Strain types were determined by sequencing the region indicated by "ST". Numbers indicate base pairs relative to the start codon of *mgpB* and *mgpC*.

and Visit 3 (Table 1). Variable regions in *mgpB* and *mgpC* were similarly PCR-amplified using the primers indicated in Fig 1 and Table 2, cloned, and sequenced from multiple plasmids. Each variable region was PCR-amplified twice, using a different primer pair for each reaction. Sequences were aligned using MultAlin (http://multalin.toulouse.inra.fr/multalin/) [41], Highlighter (https://www.hiv.lanl.gov/content/sequence/HIGHLIGHT/highlighter_top.html) [42], and ElimDupes (https://www.hiv.lanl.gov/content/sequence/elimdupesv2/elimdupes.html). Unique sequences have been deposited in GenBank (Accession numbers MT439353 – MT439593, Table 1). Sequences of the conserved regions of *mgpB* for these patient isolates have been published previously [27].

**Table 2. Primers used in this study[a].**

| Primer name | Primer sequence (5'-3') | Region amplified |
|---|---|---|
| ModPetF | GTGATGTTGTTAGTGATTGTGTG | *mgpB* strain typing region and variable region B |
| 1415R | TGGTGGTAAACATCTTAGTAGCAT | |
| MgPa-355F[b] | GAGAAATACCTTGATGGTCAGCAA | *mgpB* region B |
| 1976R | TAACTGTCAAGCATACAAACCAC | |
| 1826F | TCCAAGATGAAATGGGCAGT | *mgpB* region EF |
| 3028R | TCATTGATTACAACAAGATTACC | |
| 1653F | AGCAGGAACACTAACAATG | *mgpB* region EF |
| 3220R | GATCTCACAGTGATTTAGG | |
| 2863F | GGGAGGTGAATGGGTTGTAT | *mgpB* region G |
| 70R | CTAGTGCTAATGGTAGAAAGGG | |
| 3057F | CTTTGGGTTTCAACTTGGTG | *mgpB* region G |
| 4300R | TTGTTTTACTGGAGGTTTTG | |
| 106F | AATGTTACTGCTTACACCCC | *mgpC* region KLM |
| 1726R | TAGGGAACAGGGAGGTAACG | |
| 4149F | AAAGGCATTACAAGCAGGG | *mgpC* region KLM |
| 1549R | TAAACCTAACGCATCAAAC | |

[a], Primers target sequences that are highly conserved among strains [27, 43].

[b], previously described [44].

To measure variation during Vero coculture, one strain (MEGA 1166) was subcultured twice, for a total of ten weeks of *in vitro* growth, then variation in *mgpB* region B was assessed by PCR and sequencing as described above.

### Epitope analysis

Epitopes within MgpC (amino acids 23–938) of the *M. genitalium* type strain G37 were predicted using DiscoTope 2.0 [45] and the published crystal structure, PDB 5MZ9 [29]. For simplicity, our analyses include only those epitopes with a score greater than -1.0, corresponding to 30% sensitivity and 85% specificity. The default threshold of -3.7 corresponds to 47% sensitivity and 70% specificity. To predict epitopes in patient variants, the KLM region of G37 MgpC was replaced with sequences specific for isolates 1491 and 1534, modeled with iTasser [46], and then analyzed using DiscoTope 2.0. PyMol was used to manipulate models of predicted protein structures and generate images. Linear epitopes were predicted using Bepipred 2.0 [47].

### Ethics statement

The *M. genitalium* cultures and sera analyzed in this study were obtained from men enrolled in our Seattle-based treatment trial [35]. This study was approved by the University of Washington Institutional Review Board and all enrollees gave written informed consent.

## Results

The goal of this study was to determine the extent of sequence variation in *mgpB* and *mgpC* over time in men with NGU who were persistently infected with *M. genitalium*. Previous studies of antigenic variation of *M. genitalium* from clinical specimens have been limited to analyzing a single variable region and/or by a low number of cloned sequences analyzed [22, 23, 32, 43], which would underestimate the diversity and complexity of MgpB and MgpC variation. An appreciation of the extent of variation occurring during infection is necessary to understand how antigenic variation contributes to persistence and immune evasion. Here we analyzed sequence variation in all four variable regions (B, EF, G, and KLM) of the *mgpBC* expression site in *M. genitalium* cultured from the urine of four men with NGU during persistent infection spanning 28 to 50 days.

### Identification of suitable specimens

The *M. genitalium* isolates sequenced in this study were cultured from men enrolled in our study comparing the efficacy of doxycycline and azithromycin for the treatment of NGU [35]. Urine specimens were collected from men at multiple time points (up to four clinic visits) spanning 28 to 50 days, and confirmed *M. genitalium*-positive by PCR [48]. Strain typing [37] was used to determine that a single strain was detected at all time points. Patient sera collected at Visit 1 and Visit 3 were assayed by immunoblot to identify patients with anti-MgpB and anti-MgpC antibody reactivity. Four patients were chosen for further analysis including two (patients 10378 and 10467) with increased MgpB and MgpC reactivity between clinic visits, and two others (patients 10366 and 10477) that reacted with MgpB and MgpC at both time points (Fig 2). Immunoblot reactivity is assumed to primarily reflect the binding of patient antibodies to conserved sequences in MgpB and MgpC as variable region sequences differ between patient isolates and the G37 type strain used as antigen. The paired Visit 1 and Visit 3 *M. genitalium* isolates cultured from these four PCR-positive men were strain typed to confirm

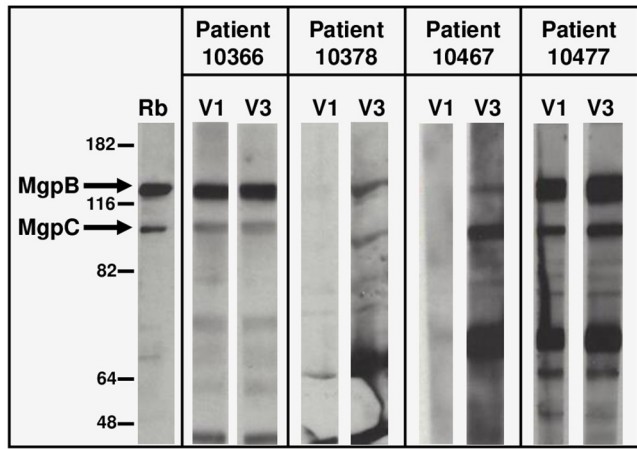

**Fig 2. Immunoblot IgG reactivity of *M. genitalium* positive patient sera with whole lysates of strain G37.** Sera obtained from four PCR-positive men at first and third clinic visits (V1 and V3, respectively), were diluted 1:1,000 and reacted with *M. genitalium* whole cell lysates separated on a 7.5% SDS-PAGE gel. "Rb", specific rabbit sera [40] was used to identify MgpB and MgpC protein bands (arrows). Molecular weight markers (in kDa) are shown at left. The magnitude of antibody reactivity between patients cannot be compared as different film exposure times were used.

that the Vero-cultured isolates were identical to the strain present in patient specimens (Table 1). These isolates were then analyzed for sequence variation over time.

## Analysis of sequence diversity in *M. genitalium* infected men

To assess gene variation in *M. genitalium* isolates, each of the three variable regions within *mgpB* (regions B, EF, and G) and the single variable region within *mgpC* (region KLM) was PCR amplified from Visit 1 and Visit 3 cultures (Fig 1). Amplicons were cloned, and then 10 plasmids were sequenced to identify regions that varied between time points. If sequence changes were observed in a particular variable region, then an additional 25–30 plasmids were sequenced. These variable regions were then amplified with a second primer pair targeting the same region (Fig 1 and Table 2), again cloning and sequencing 35–40 plasmids. Similar sequences were obtained using both primer pairs suggesting that each reaction amplified representative sequences. For example, 18 variant sequences were identified among 78 clones of region B from Patient 10366 Visit 1. Of the 6 variant sequences found in more than two plasmid clones, all were amplified by both primer pairs, representing 83% of total sequences obtained. Using this strategy approximately 75 cloned sequences were analyzed per variable region for each time point in all four patients.

The predicted amino acid sequences for each variable region were aligned to assess sequence changes between time points. Sequence variation between time points was observed in all four patients analyzed in at least one variable region. Variable region sequences were unique to the isolates from each patient (i.e., no sequence was identical between different patients), emphasizing the diversity of *M. genitalium* strains circulating in a single geographic area. A comparison of the Visit 1 and Visit 3 sequences revealed the loss of specific amino acid sequences over time, consistent with immune selection against specific epitopes. Figs 3 and 4 show the results of these analyses in graphic form in which each individual cloned sequence is compared to the predominant sequence present at Visit 1. *M. genitalium* isolates cultured from Patient 10366 varied between time points in all four regions of *mgpB* and *mgpC* (Fig 3). A mixture of variant region B, EF, and KLM sequences, with a single region G sequence, was present at Visit 1. However, by Visit 3 novel sequences predominated in all four variable regions

## Patient 10366

**Fig 3. Evolution of MgpB regions B, EF, and G, and MgpC region KLM sequences during infection in Patient 10366.** Amino acid alignments (Multalin [41]) were submitted to Highlighter [42] to generate the output shown. Each horizontal line represents a single cloned sequence. Amino acids that differ from the predominant Visit 1 sequence (top line) are marked with vertical colored bars with different colors corresponding to particular amino acids. Visit 1 and Visit 3 sequences are indicated by grey and yellow block shading at right, respectively. Dashed boxes indicate sequences detailed in Figs 5 and 6. Variable regions not shown for Patients 10378, 10467, and 10477 did not vary substantially between time points and therefore were not analyzed further.

consistent with selection against sequences present at Visit 1. In contrast, Patient 10467 varied only in Region B, Patient 10378 varied only in region EF, and Patient 10477 varied in Regions EF and KLM (Fig 4). In general, a single variant sequence predominated at Visit 3 for all four patients analyzed, although patients differed in whether they were infected with a variety of variants (eg Patient 10366 B, EF, and KLM) or a single predominant sequence (eg Patients 10378, 10467, and 10477). Interestingly, in Patient 10378, the Visit 3 culture was a mixture of the predominant Visit 1 sequence and a novel sequence (Fig 4). This may indicate "selection in process"–i.e., that effective antibodies have recently appeared and are actively selecting against an epitope formed by the amino acids that have changed between time points.

Close inspection of variant sequences (Figs 5 and 6) revealed that changes occurred in clusters of amino acids consistent with the well-described mechanism of segmental recombination between *mgpBC* and the Mgpars [22, 23]. As individual variant clusters arose independently of each other (for example, amino acids "DTSG.T" appeared independently of amino acids "KSG" in region B Patient 10366, Fig 5) we assumed that they represent independent

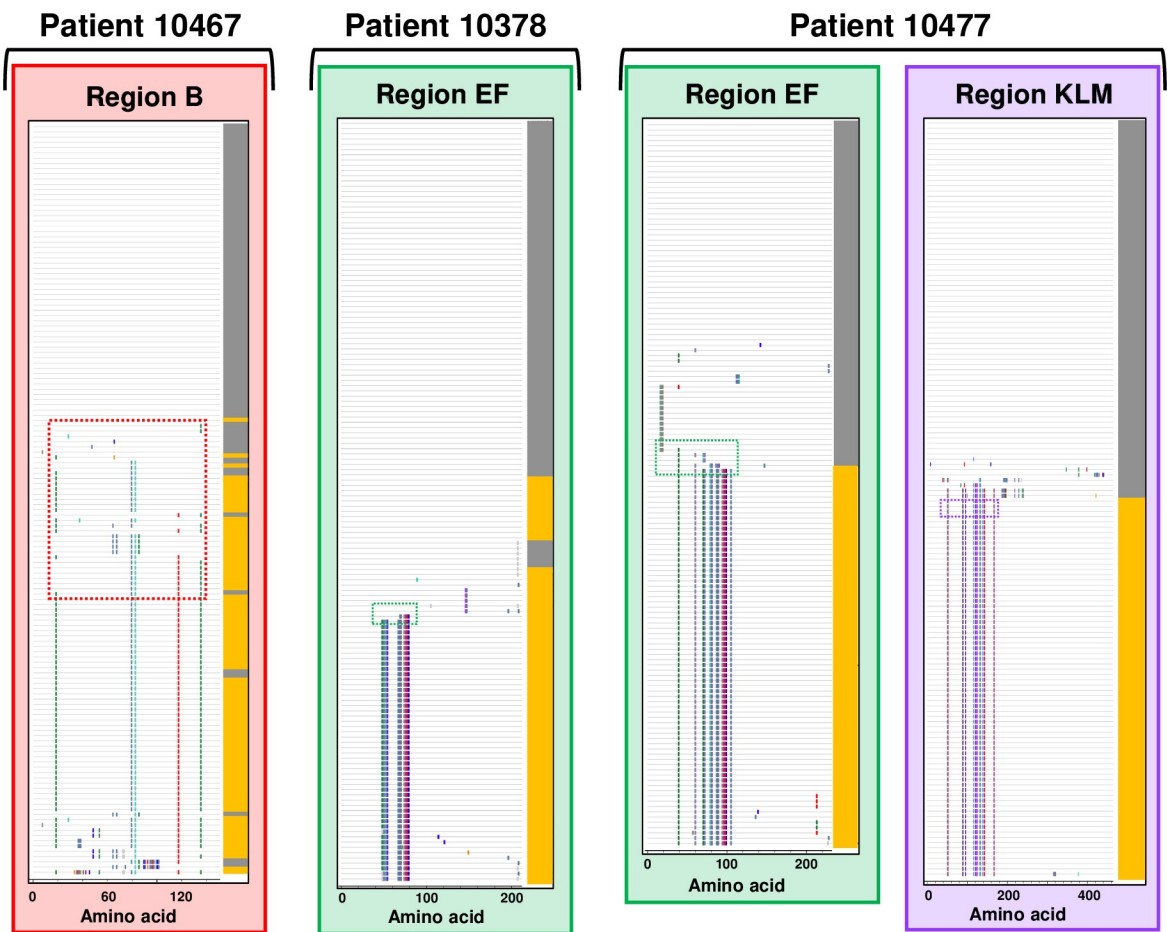

**Fig 4. Evolution of MgpB regions B, EF, and G, and MgpC region KLM sequences during infection in Patients 10378, 10467, and 10477.** Analysis and data presentation are described in legend for Fig 3.

recombination events. To estimate the number of recombination events in these patients, we visually inspected the amino acid alignments shown in Figs 3 and 4 for segmental sequence changes, similar to previously described analyses [49]. For simplicity, single amino acid changes present in only one plasmid clone were omitted as these could arise via point mutations or PCR/sequencing errors. This analysis identified 68 possible recombination events among the variants infecting Patient 10366 (8 in region B, 18 in EF, 1 in region G, and 41 in region KLM); 6 recombination events in Patient 10378 (region EF only), 16 in Patient 10467 (region B only), and 26 in Patient 10477 (11 in region EF and 15 in KLM).

## Stability of sequences in Vero coculture

As recovery of *M. genitalium* from patient specimens requires three weeks of *in vitro* coculture with Vero cells, we considered the possibility that the gene variation we observed occurred during *in vitro* culture rather during patient infection. To address this issue, we serially passaged *M. genitalium* from Patient 10366 at Visit 1 (MEGA 1166) for an additional seven weeks in Vero cells then compared the variants present in MgpB region B to the same unpassaged culture. As shown in Fig 7 (upper panel), we found that the number of different variants present, and the sequences of these variants, changed very little during these seven weeks of *in*

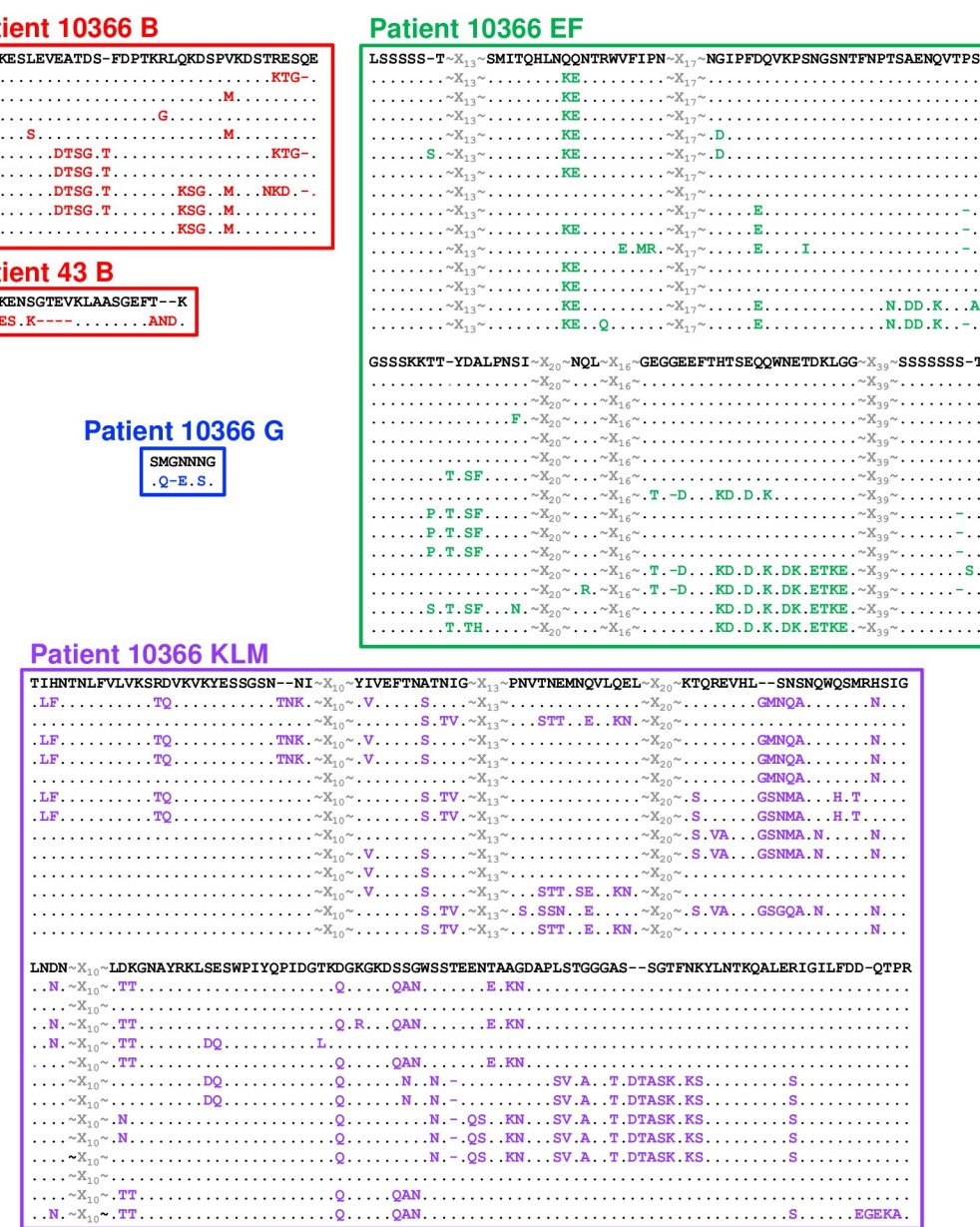

**Fig 5. Detailed alignment of variable segments indicated in Fig 3 for Patient 10366.** The predominant sequence present at Visit 1 is shown in black at the top of each alignment. Each unique sequence is represented on a single line, variant amino acids are marked in color, dots indicate unchanged amino acids, and dashes indicate gaps.

*vitro* growth. Similarly, *M. genitalium* cultured from this patient at Visit 3 (MEGA 1206) varied little during seven weeks *in vitro* (Fig 7, lower panel). These results contrast with the extensive sequence evolution that occurred during 33 days of urethral infection and support our conclusion that diversification of sequences is a consequence of growth *in vivo*.

## MgpC variant amino acids lie within predicted conformational B cell epitopes

The recent description of the MgpC crystal structure [29] afforded the opportunity to predict the location of conformational B cell epitopes. As shown in Fig 8A, DiscoTope [45] analysis

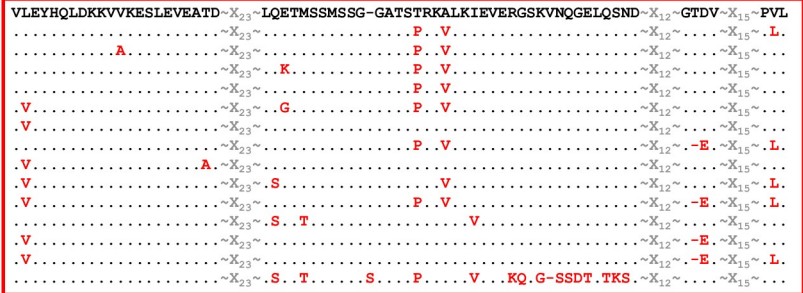

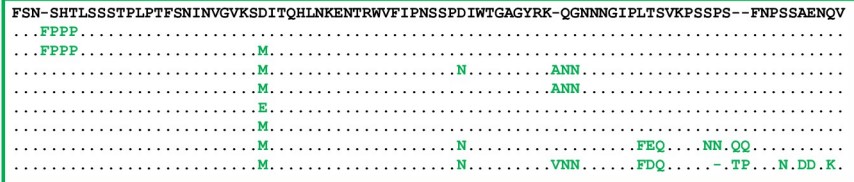

**Fig 6. Detailed alignment of variable segments indicated in Fig 4 for Patients 10378, 10467, and 10477.** The predominant sequence present at Visit 1 is shown in black at the top of each alignment. Each unique sequence is represented on a single line, variant amino acids are marked in color, dots indicate unchanged amino acids, and dashes indicate gaps.

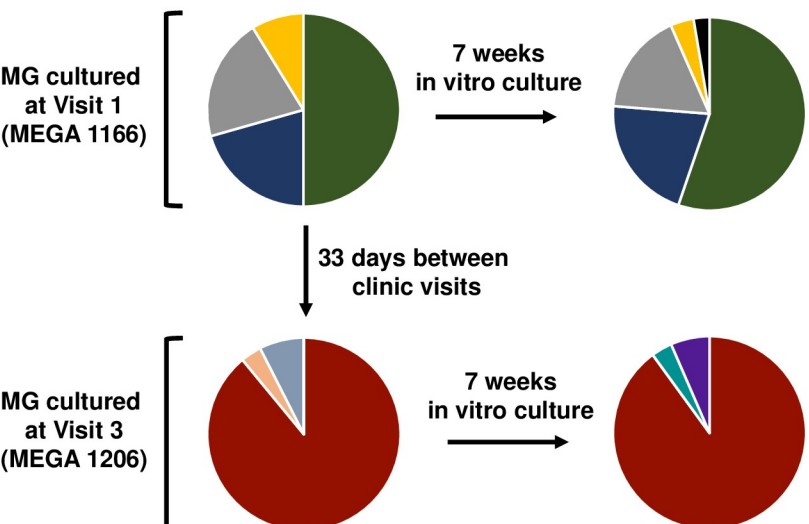

**Fig 7. Assessment of antigenic variation during in vitro passage in Vero cell cocultures.** MgpB region B sequences were compared between Visit 1 (MEGA 1166) and Visit 3 (MEGA 1206) cultured from patient specimens after seven weeks of in vitro passage in Vero cell cocultures. Identical sequences are indicated with identical colors; 34–38 plasmids were sequenced from each culture. Results show that little variation occurred during seven weeks of in vitro growth for each culture.

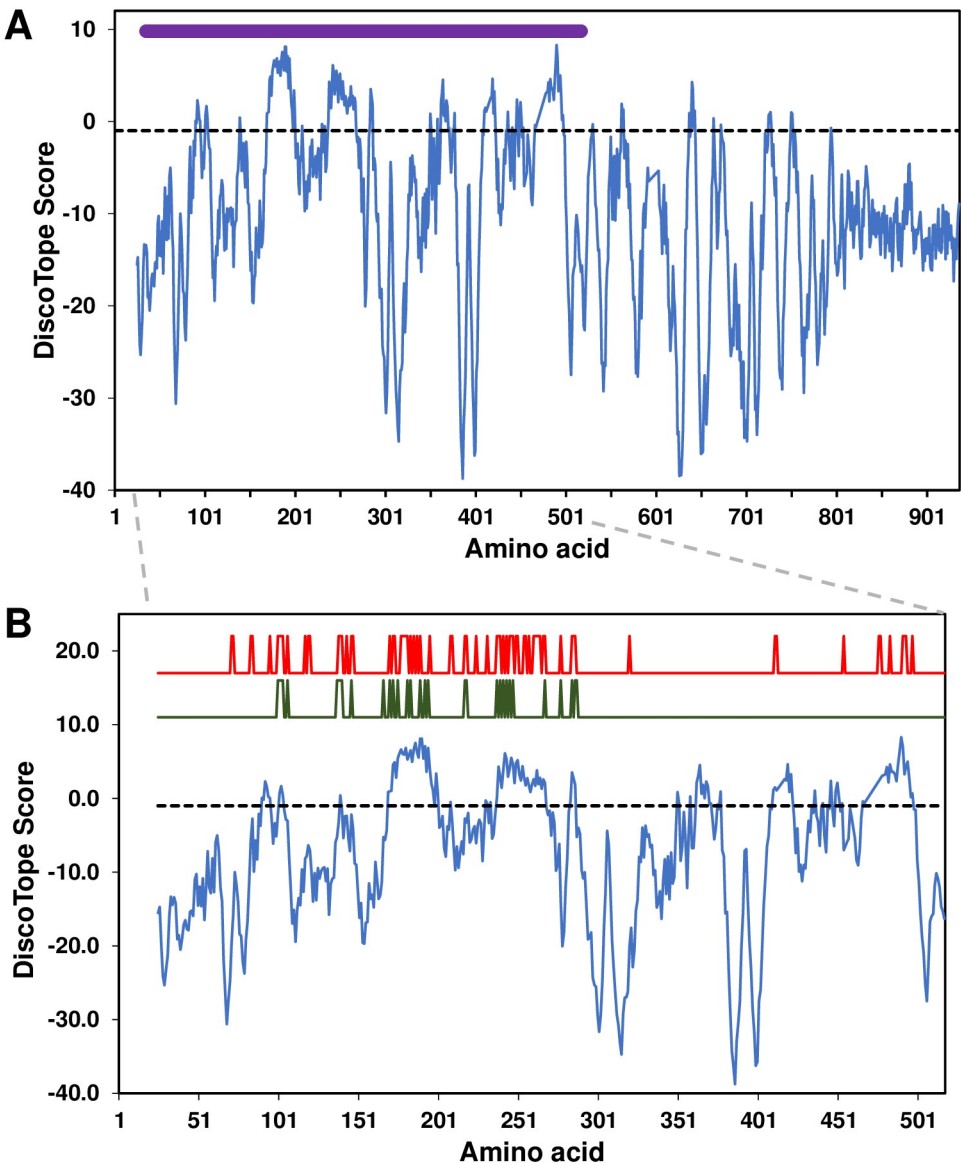

**Fig 8. Prediction of conformational B cell epitopes in MgpC. (A).** G37 MgpC high-scoring B cell epitopes localize to variable region KLM. DiscoTope 2.0 with a stringent cutoff of -1 (corresponding to 30% sensitivity and 85% specificity), indicated by the horizontal dashed line, was used to predict conformational B cell epitopes using the published MgpC structure (PDB 5mzb, amino acids 25–936 [29]). Variable region KLM is indicated by the purple line, this region is expanded to show detail in panel B (dashed grey lines). **(B).** Variant amino acids correlate with predicted B cell epitopes within variable region KLM. Red and green lines indicate sequences obtained for Patient 10366 and 10477 isolates, respectively, amino acids that varied between time points indicated by "peaks" (arbitrary units).

predicted numerous conformational epitopes within full-length MgpC with higher scoring epitopes concentrated in variable region KLM: 152 amino acids in full-length MgpC have a DiscoTope score greater than -1.0 (corresponding to 30% sensitivity and 85% specificity), 133 (87.5%) of which are located in KLM.

The locations of predicted conformational epitopes were mapped onto the crystal structure of G37 MgpC as shown in Fig 9. Epitopes predicted within the conserved region of MgpC are marked in red, while epitopes in variable region KLM that cluster together on the MgpC

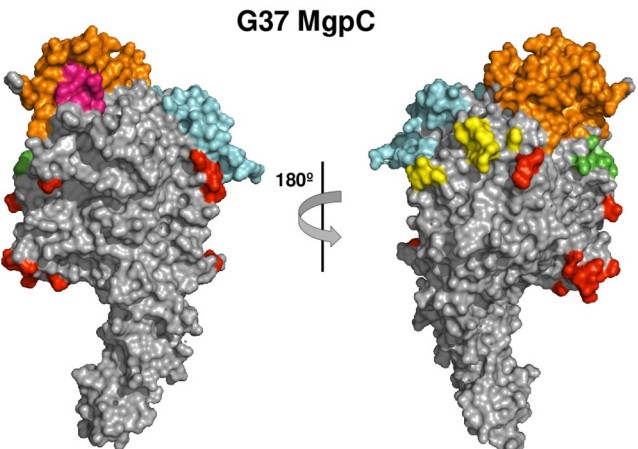

**Fig 9. Location of predicted conformational epitopes on the G37 MgpC protein.** The MgpC transmembrane domain (not shown) is oriented downward. Colored residues indicate epitopes that group together on the surface of MgpC; epitopes in the conserved region of KLM are indicated in red. The eight amino acids required for sialic acid binding [29] are shown in magenta, five of which lie within predicted epitopes. Figure produced using Pymol.

surface are indicated with various colors. The amino acids implicated in sialic acid binding [29] are located within, or adjacent to, predicted epitopes (magenta in Fig 9). This analysis shows that most conformational epitopes are located on the so-called "crown" of MgpC, which consists primarily of variable region KLM, as previously noted [29], and are located on the surface of the MgpC molecule where they could be targeted by host antibodies.

## Variation observed during patient infection localizes to predicted epitopes

We hypothesized that if variation in MgpC region KLM represents an immune evasion mechanism then sequence changes should affect predicted epitopes. We aligned the KLM sequences obtained from two patients and identified amino acids that changed between time points (Fig 8B). In Patient 10366, 32 amino acids varied between early and late time points in region KLM, 22 (66%) of which corresponded to epitopes predicted by DiscoTope with a stringent cutoff of -1.0 (indicated as peaks in Fig 8B, red line). Similarly, 49 (67%) of the 73 amino acids that varied in Patient 10477 were located in epitopes (Fig 8B, green line).

We next determined whether sequence variation *in vivo* changed the amino acid sequence of an existing epitope, or if a pre-existing epitope was changed to a non-epitope. For this analysis, we replaced the G37 MgpC region KLM sequence with the patient-specific variant sequences, generated structural models with iTasser, and predicted conformational epitopes with DiscoTope. For simplicity, sequences from a single patient (Patient 10477: Visit 1 MEGA 1491 vs Visit 3 MEGA 1534) were analyzed as a single, unique sequence predominated at each time point (Fig 10). The epitopes predicted using the patient-specific models were similar to G37 in score (not shown) and location (compare Figs 10 to 9) despite 17% amino acid sequence differences. Patient-specific epitopes localized predominantly to the crown of MgpC and the amino acids that changed between time points (indicated in black in Fig 10) were embedded in these epitopes. Interestingly, the variant amino acids localized to different faces of the MgpC crown suggesting that several epitopes changed during the course of infection, possibly avoiding binding by antibodies of multiple specificities.

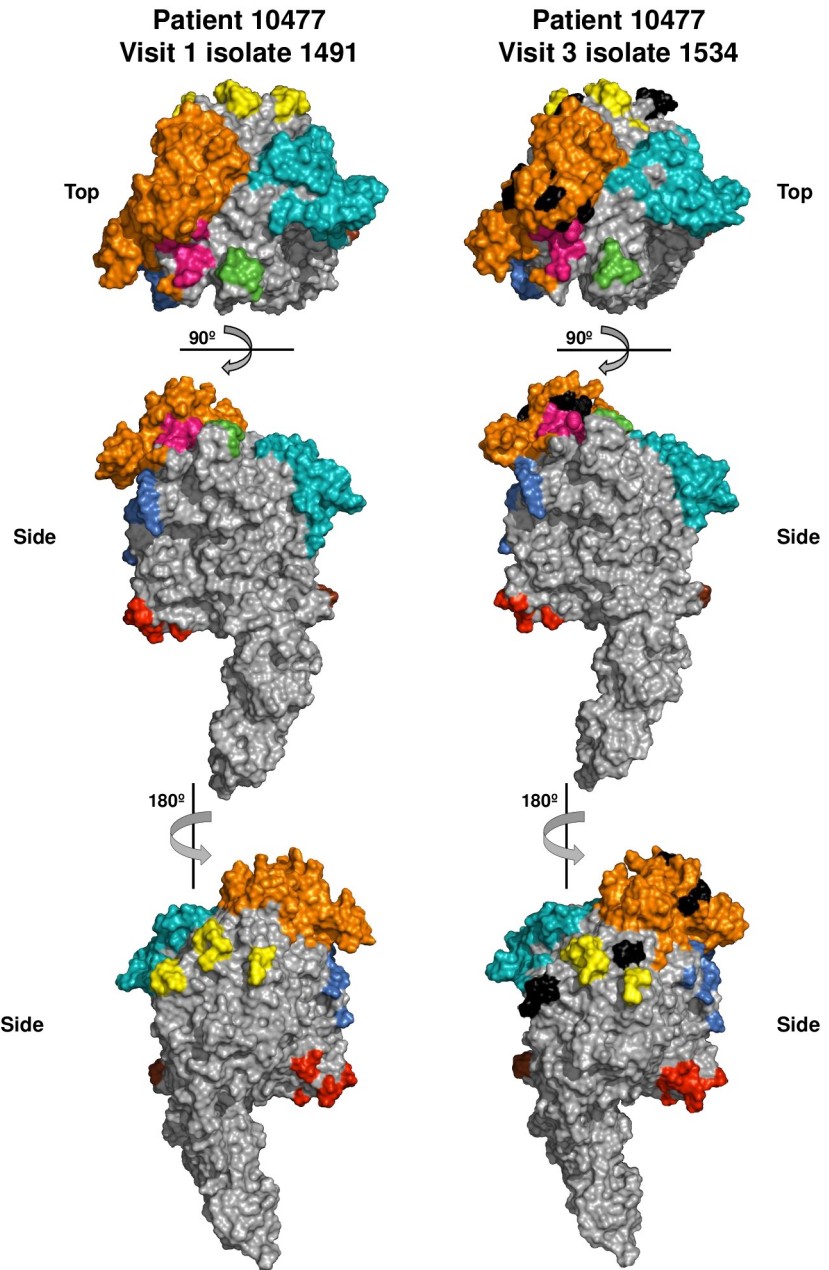

**Patient 10477
Visit 1 isolate 1491**

**Patient 10477
Visit 3 isolate 1534**

**Fig 10. Models of MgpC for Patient 10477 Visit 1 isolate (MEGA 1491, left) and Visit 3 isolate (MEGA 1534, right).** Colored residues indicate predicted epitopes that group together on the surface of MgpC, added to improve data visualization. Amino acids that changed between time points are indicated in black on the Visit 3 model. Amino acids implicated in sialic acid binding are shown in magenta. Residues indicated in red and brown are epitopes predicted in the conserved region of MgpC.

## Discussion

In this study, we assessed sequence variation in the variable regions of *mgpB* (regions B, EF, and G) and *mgpC* (region KLM) of *M. genitalium* cultured from longitudinal specimens (spanning 30 to 58 days) from persistently infected men. Strain typing confirmed that a single strain type was detected in each patient at all time points, and immunoblots indicated antibody

reactivity to the MgpB and MgpC proteins in the sera of all four men. Evidence of extensive variation was observed in these patients in one or multiple *mgpBC* variable regions *in vivo*, with little variation during *in vitro* culture. The extent and diversity of variation was greater than previously appreciated by other studies of *M. genitalium* antigenic variation in patient specimens. Most variable amino acids in MgpC mapped to predicted conformational B cell epitopes supporting a role for antigenic variation as a mechanism to avoid the biologic effect of specific antibodies in order to persist *in vivo*.

Previous studies have been instrumental in establishing that the gene variation predicted from *in vitro* studies occurs *in vivo*. For example, we previously observed *mgpB* and *mgpC* gene variation in *M. genitalium*-infected women [22, 23] and Ma *et al.* [32, 43] documented variation in several *M. genitalium*-infected men and women. However, each of these previous studies assessed a limited number of sequences (5 to 18 cloned sequences per specimen) and only one or two variable regions in these patients. Fookes *et al.* [50] assessed changes in the entire *mgpBC* operon by whole genome sequencing of isolates obtained 79 days apart, however, as single colony cloned strains were sequenced, the full breadth of variation within the infecting population could not be assessed. Our study provides a comprehensive assessment of variation across all variable regions in both MgpB and MgpC in men who were each persistently infected with a single strain. This approach allows an appreciation of the extent and frequency of variation, necessary to understand how antigenic variation relates to immune evasion and pathogenesis. Interestingly, we found that some variable regions did not change between time points in three patients: few changes were observed in regions B, G, and KLM in Patient 10378, in regions EF, G, and KLM in Patient 10467, and in regions B and G in Patient 10477. Our model of antibody-mediated immune selection would predict that the sera of these patients would react poorly to these non-variant regions of MgpB and MgpC, a prediction we intend to test in future experiments.

We compared the number of recombination events observed at the expression site, calculated by identifying clusters of amino acid changes between time points and assuming that a unique cluster arose via a single recombination event. In this small sample set, the number of recombination events did not correlate with the length of time between specimen collection. For example, in Patient 10378 we observed 6 recombination events over 48 days of infection, whereas 68 recombination events were detected in Patient 10366 during 33 days of infection. It is tempting to speculate that the duration of antibody response is related to the number of variants observed. For example, isolates from Patients 10366 and 10477 had the most recombination events between time points analyzed (68 and 26, respectively) and both patients had anti-MgpB and MgpC antibodies at early and late time points, providing more opportunity for antibody-mediated selection. However, it is unknown when antibodies arose in Patients 10378 (48 days between visits) and 10467 (28 days). Furthermore, patient serum antibody reactivity was measured by immunoblot against whole cell lysates of *M. genitalium* strain G37, thereby detecting reactivity to sequences common in all MgpB and MgpC alleles, rather than unique variants present at early time points in these patients. Finally, immunoblot reactivity does not necessarily indicate biologic activity, for example, antibodies may target epitopes that are not exposed on the surface of *M. genitalium*. Further experiments are needed to ascertain whether these patient antibodies bind specific variants and have biologic activity.

Our results showed clearly that many more variants arise during patient infection than during *in vitro* culture, probably due to a combination of immune selection and higher rates of recombination *in vivo*. The role of selection by immune factors is supported by the presence of antibodies to MgpB and MgpC, the known immunogenicity of these proteins in humans and animals, and the prediction of many high scoring conformational B cell epitopes in the variable region of MgpC. Furthermore, the fact that a single variant predominates at a given time

point suggests that immune selection drives variation (ie, by selecting against one sequence followed by proliferation of a novel variant that escapes antibody selection). In concert with immune selection, we hypothesize that the rate of recombination is upregulated *in vivo*. Recently, an alternative sigma factor, MG428, was identified that induces expression of RecA and other recombination enzymes thereby increasing antigenic and phase variation [51–53]. We hypothesize that specific inducing signals for the MG428 regulon, as yet unknown, will be found in the genital tract.

We analyzed the MgpC protein for predicted conformational B cell epitopes using the available crystal structure [29]. We found that most conformational epitopes are located in the variable KLM region of MgpC and that sequence variation detected in *M. genitalium* patient isolates alters the amino acids specifically localized to these epitopes. These data support our hypothesis that the role of gene variation is avoidance of specific antibody. Interestingly, few conformational epitopes are predicted in conserved sequences of MgpC suggesting that low immunogenicity may be an additional strategy to avoid antibody targeting of these invariant yet surface exposed regions. Further studies are needed to determine if epitopes predicted *in silico* are indeed targeted by host antibodies, and if the MG281 antibody binding protein of *M. genitalium* [54] plays a role in immune evasion.

## Supporting information

**S1 Fig. Unadjusted immunoblot image.** Original blots showing reactivity of patient sera to *M. genitalium* whole cell lysates.
(TIF)

## Author Contributions

**Conceptualization:** Gwendolyn E. Wood, Stefanie L. Iverson-Cabral, Patricia A. Totten.

**Data curation:** Catherine W. Gillespie.

**Formal analysis:** Gwendolyn E. Wood, Patricia A. Totten.

**Funding acquisition:** Gwendolyn E. Wood, Lisa E. Manhart.

**Investigation:** Gwendolyn E. Wood, Stefanie L. Iverson-Cabral, Catherine W. Gillespie, M. Sylvan Lowens.

**Methodology:** Gwendolyn E. Wood.

**Project administration:** Lisa E. Manhart, Patricia A. Totten.

**Visualization:** Gwendolyn E. Wood.

**Writing – original draft:** Gwendolyn E. Wood.

**Writing – review & editing:** Gwendolyn E. Wood, Stefanie L. Iverson-Cabral, Catherine W. Gillespie, Lisa E. Manhart, Patricia A. Totten.

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
