## [Decision Letter · Decision Letter 0]

28 Aug 2020

PONE-D-20-17903

Sequence variation and immunogenicity of the Mycoplasma genitalium MgpB and MgpC adherence proteins during persistent infection of men with non-gonococcal urethritis

PLOS ONE

Dear Dr. Wood,

Thank you for submitting your manuscript to PLOS ONE. I want to apologize for the delay- finding reviewers was challenging this summer. After careful consideration, we feel that it has merit but does not fully meet PLOS ONE’s publication criteria as it currently stands. Therefore, we invite you to submit a revised version of the manuscript that addresses the points raised during the review process.

We look forward to receiving your revised manuscript.

Kind regards,

Catherine A. Brissette, Ph.D.

Academic Editor

PLOS ONE

Journal Requirements:

Reviewers' comments:

Reviewer's Responses to Questions

**Comments to the Author**

1. Is the manuscript technically sound, and do the data support the conclusions?

Reviewer #1: Yes

Reviewer #2: Yes

Reviewer #3: Partly

2. Has the statistical analysis been performed appropriately and rigorously? 

Reviewer #1: Yes

Reviewer #2: Yes

Reviewer #3: N/A

3. Have the authors made all data underlying the findings in their manuscript fully available?

Reviewer #1: Yes

Reviewer #2: Yes

Reviewer #3: Yes

4. Is the manuscript presented in an intelligible fashion and written in standard English?

Reviewer #1: Yes

Reviewer #2: Yes

Reviewer #3: Yes

5. Review Comments to the Author

Reviewer #1: The manuscript by Wood and colleagues reports on the sequence variation of two highly variable adhesion related proteins of the STI pathogen, Mycoplasma genitalium. This species uses rampant recombination between variant sequences to shuffle new coding sequences into a single expression locus. Although the main features of the process have been known for a while, this study documents variation seen isolates from four persistently colonized men. Overall the data are quite clear cut, although it was no unambiguously clear, how much of the MgpB variation was previously reported in reference 27 (please see below)

Specific Points

1. Introduction. Since there are a number of different protein names used for MgpB and MgpC, it would be informative to include these in parentheses the first time the proteins are introduced (e.g. L57; for example include MG numbers and P110 etc).

2. L68 The term MgPars is introduced here, without any information provided about their nature.

3. L70 It is not clear how recombination mediated antigenic variation and the “exchange of large regions of mgpBC with single or multiple MgPars” i.e. phase variation, are actually different processes. Is it the length of the region shuffled? In which case this third mechanism of phase variation would seem to be one extreme end of the antigenic variation recombination spectrum. This distinction was unclear when I read it.

4. L71 It is not clear how recA (the primary mechanism for these phenomena) can generate point mutations (L70) unless some of the MgPars carry such isolated mutations.

5. L83. This might be interpreted that immune selection is causing the variation directly, whereas presumably it is the driver of selecting against the previous antigenic epitopes.

6. L134. Please include the name of the PCR polymerase (hi fidelity? ) and the typical number of cycles that were necessary to amplify the variable regions. These points are important when trying to assess the sequence variation obtained.

7. L30 and L135. As you read the methods, there are day numbers that seem discrepant. On L30, DNA is prepared from co-cultured cells after day 7, 14, 21, 28 but then on L135 the templates are from three or 10 weeks.

8. L147. The authors state that the conserved regions of mgpB for these isolates were reported in ref 27. Table S3 from that paper seems to include variable regions also, so it is not 100% clear what was reported before and what is new (in regards to variable sequences in MgpB).

9. L189 and ref 37. How many different strain types are known? And how many different strains can have the same “strain type”? Although it is possible that the strain typing shows that a “single strain across all time points” was present, it does not categorically prove that. It would be more accurate to state that only a single strain type was detected. Also L352 “with a single persistent strain type…”?

10. Most of the sequence analysis comes from clones of PCR products obtained from co-cultured cells and uses the 3 week DNA prep. Was insufficient DNA recoverable at earlier times (the methods indicate that day 7 and day 14 DNA preps were made, but these were not characterized)? What was the rationale for this?

11. Is the recombination reciprocal and do the authors have data from any of the MgPar sites?

Minor points

1. L47 Should “Chlamydia” be italicized?

2. L99 versus L121. There is variable hyphenation usage in “co-culture”. Please use one style throughout.

3. L119, L124, L132, L137. Please provide ECL and pCR2.1 supplier details and check to see if company locations are required for ATCC, Epicentre and PCR enzyme (see point 6 above)

4. Figure 2-indicate what he “Rb” refers to in the legend.

5. L402 i.e. (should this be italicized)?

6. L416 should “in silico” be italicized?

Reviewer #2: The main goal of the study from Wood et al., is to determine the extent of sequence variation in the MgpB and MgpC over time in men persistently infected with Mycoplasma genitalium. MgpB and MgpC are adherence proteins, which are immunodominant targets of host antibodies.

The work extends previous studies (analyzing mostly a single variable region or with other limitations) carried out by some of the coauthors and by other groups. The methodology used and the analysis performed in the present work seem appropriated and the conclusions consistent. However, the amount of patients used (four) in the study is maybe a bit too small and the novelty of results limited.

It would be nice to indicate at some point in the text the alternative nomenclature of P140 and P110 for MgpB and MgpC, respectively.

In the work the analysis is performed in parallel for MgpB and MgpC, except for the mapping of the variant amino acids onto the protein structure that is only done for MgpC. Since some weeks ago the structure of MgpB was published and it is now also available. In my opinion the mapping of variants should now also be done for MgpB and included in this work.

Reviewer #3: In this manuscript, Wood and colleagues examine the generation of sequence variation within the main adhesins of M. genitalium during persistent infection. This variation is intended to modify/replace the antigenic regions of the MgpB and MgpC adhesins to avoid host antibody recognition. In contrast to previous studies, where the analysis was limited to a single region or a low number of cloned sequences, the current report addresses sequence variation in all four variable regions of the mgpBC expression site and analyze multiple clones. Therefore, the present study provides a more complete picture of the diversity and complexity of MgpB and MgpC variation over time.

The topic of study is critical to understand persistence of M. genitalium in the genital tract and the authors have wide experience investigating sequence variation in this pathogen. Overall, the results are well-presented and are easy to follow, so I do not have specific comments on this regard. However, in my opinion, there are a few questions regarding the study design that would need to be clarified:

1) To identify sequence variants (L134-141 and L207-217), the authors use PCR amplification, followed by cloning and Sanger sequencing of multiple plasmids (around 75). The number of individual clones analyzed seems sufficient to obtain a representative view of the predominant sequence variants present at each time point. However, clone selection is always a matter of concern. Therefore, could Next Generation Sequencing technology have been used to analyze the different PCR amplicons? If so, do the authors think that the use of NGS could impact or improve the results presented in this study? Could the use of NGS benefit future studies addressing sequence variation of Mge during persistent infection?

2) The Mge isolates analyzed were recovered by co-culture of processed patient urine with Vero cells (L120-133). The possible impact of co-culture with Vero cells (three or more weeks) on sequence variation is also matter of concern (L272-282). In this regard, could this step introduce some bias in the sequence variants identified? For example, selecting for adhesin variants with increased adherence capacity or some other feature important for in vitro survival. Therefore, was this co-culture strictly necessary? Can DNA from Mge be extracted/sequenced directly from a patient sample (for example the urine pellets after centrifugation)? In addition, why was only culture supernatants used to isolate DNA (L131)? I do not have experience isolating Mge clinical strains but, shouldn’t the bacteria be attached to Vero cells?

3) The current study seems appropriate to determine the extent of sequence variation in the adhesin genes over time in men with NGU persistently infected with Mge (L174). However, the relationship between the observed changes and antigenic variation, and the impact on antibody recognition in less apparent. In this sense, all evidences are indirect. Even if amino acid changes localize (in general) to regions predicted as epitopes, no direct evidence is provided that these changes effectively avoid antibody recognition. Therefore, all conclusions on this regard must be limited and presented as a possibility rather than a fact.

To establish a solid relationship between sequence variation and antibody recognition, direct evidence is required. For example, for each particular Mge isolate, is it possible to test if patient serum from Visit 1 is less or non-reactive against MgpB or MgpC variants predominantly expressed at Visit 3? To test this, can the Mge strains expressing the adhesin variants predominant at Visit 3 be isolated and used as template (whole cell lysates) for western blotting analysis using sera from Visit 1? Alternatively, is it possible to generate Mge mutants expressing these new variants that could be analyzed by western blotting instead of G37 whole cell lysates? Moreover, could the predominant variants (MgpB/MgpC) identified at Visit 3 be expressed as recombinant proteins (heterologous expression) and used to demonstrate loss of recognition by serum from Visit 1?

Minor points:

Why are Figure legends embedded within the manuscript?

L64 “is affected by two mechanisms”: rather than two mechanisms, I think antigenic and phase variation are two processes or even purposes.

L99 “Among the four”. New paragraph.

L135 “three (or ten weeks)”. When ten?

L207-217. This is Methods rather than Results?

L224 “consistent with immune selection against specific epitopes”. Could sequence variation have other purposes? Perhaps not all sequence changes are aimed to avoid immune system recognition and are negatively/positively selected by other forces with different purposes.

L226 “the predominant sequence”. Are “predominant sequences” much more abundant than other sequences in each sample? Can the authors show the % of identification of the predominant sequence for each isolate and time point? Were mixtures often identified (other than for patient 43 Visit 1)?

6. PLOS authors have the option to publish the peer review history of their article (what does this mean?). If published, this will include your full peer review and any attached files.

Reviewer #1: No

Reviewer #2: No

Reviewer #3: No

---

## [Author Response · Author response to Decision Letter 0]

28 Sep 2020

Response to Reviewers

PONE-D-20-17903

Sequence variation and immunogenicity of the Mycoplasma genitalium MgpB and MgpC adherence proteins during persistent infection of men with non-gonococcal urethritis

We thank the reviewers for their thorough critique of our manuscript. We have incorporated their suggestions as outlined below.

Editor comment regarding data not shown: 

Response: We have briefly explained these data in Results.

Reviewer 1

1. Introduction. Since there are a number of different protein names used for MgpB and MgpC, it would be informative to include these in parentheses the first time the proteins are introduced (e.g. L57; for example include MG numbers and P110 etc).

Response: These designations have been added.

2. L68 The term MgPars is introduced here, without any information provided about their nature.

Response: We have expanded the description of the MgPars.

3. L70 It is not clear how recombination mediated antigenic variation and the “exchange of large regions of mgpBC with single or multiple MgPars” i.e. phase variation, are actually different processes. Is it the length of the region shuffled? In which case this third mechanism of phase variation would seem to be one extreme end of the antigenic variation recombination spectrum. This distinction was unclear when I read it.

Response: Our previous publication (Burgos et al 2018) describes multiple ways in which recombination can produce phase variants. We have revised this paragraph in Introduction to clarify the differences between antigenic and phase variation.

4. L71 It is not clear how recA (the primary mechanism for these phenomena) can generate point mutations (L70) unless some of the MgPars carry such isolated mutations.

Response: We did not mean to imply that RecA generates point mutations, and have revised that sentence to clarify. RecA is required for most antigenic and phase variants, but a small number of variants arise in recA mutants. However, it is also true that point mutations can be found in the MgPars which prevent MgpB/C expression when recombined into the expression site.

5. L83. This might be interpreted that immune selection is causing the variation directly, whereas presumably it is the driver of selecting against the previous antigenic epitopes.

Response: We have revised this sentence.

6. L134. Please include the name of the PCR polymerase (hi fidelity? ) and the typical number of cycles that were necessary to amplify the variable regions. These points are important when trying to assess the sequence variation obtained.

Response: These details have been added. We used Platinum PCR SuperMix High Fidelity and 30 cycles of amplification.

7. L30 and L135. As you read the methods, there are day numbers that seem discrepant. On L30, DNA is prepared from co-cultured cells after day 7, 14, 21, 28 but then on L135 the templates are from three or 10 weeks.

Response: Sorry for the confusion. All strains were analyzed after three weeks in culture. One strain was further subcultured twice (for a total of ten weeks in vitro) to determine if variants arose during Vero coculture. We have clarified this in the methods.

8. L147. The authors state that the conserved regions of mgpB for these isolates were reported in ref 27. Table S3 from that paper seems to include variable regions also, so it is not 100% clear what was reported before and what is new (in regards to variable sequences in MgpB).

Response: Only the conserved sequences of mgpB and mgpC for these patient isolates are included in our prior publication. For clarity, we have changed the patient designations in the current manuscript (Patients 43, 47, 68 and 70) so that they match our previous publication (Patients 10366, 10378, 10467, and 10477, respectively).

9. L189 and ref 37. How many different strain types are known? And how many different strains can have the same “strain type”? Although it is possible that the strain typing shows that a “single strain across all time points” was present, it does not categorically prove that. It would be more accurate to state that only a single strain type was detected. Also L352 “with a single persistent strain type…”?

Response: We have modified as suggested to “only a single strain type was detected”. Dozens of M. genitalium strain types have been described, and this strain typing method has been used in several studies to confirm sexual transmission M. genitalium, track the development of antimicrobial resistance during treatment, and distinguish persistent infection from reinfection. In the particular study from which these strains were obtained, we found 26 strain types among the 46 men tested (Totten et al, manuscript in prep). In all cases of persistent NGU, the same M. genitalium strain type was detected at multiple clinic visit.

10. Most of the sequence analysis comes from clones of PCR products obtained from co-cultured cells and uses the 3 week DNA prep. Was insufficient DNA recoverable at earlier times (the methods indicate that day 7 and day 14 DNA preps were made, but these were not characterized)? What was the rationale for this?

Response: In future studies, we intend to study the effect of patient sera on viable cells of the M. genitalium strains isolated from patients at different clinic visits to determine if antibody reactivity correlates with loss of particular variants from patient specimens. We therefore determined sequences for the Vero coculture time point with the maximum number of M. genitalium cells (corresponding to late log phase) for our analysis. The weekly DNA preps were used for qPCR to confirm growth (increase in genomes) of the M. genitalium strains from patient urine. We have edited this paragraph for clarity.

11. Is the recombination reciprocal and do the authors have data from any of the MgPar sites?

Response: We have not sequenced the MgPars from these isolates to confirm whether recombination was reciprocal in these variants. In addition, as these are mixed populations it would be difficult to sort out which MgPar sequences corresponded to particular expression site sequences in a single cell. 

Minor points

1. L47 Should “Chlamydia” be italicized?

Response: We have edited to: Chlamydia trachomatis.

2. L99 versus L121. There is variable hyphenation usage in “co-culture”. Please use one style throughout.

Response: We have changed to “coculture”.

3. L119, L124, L132, L137. Please provide ECL and pCR2.1 supplier details and check to see if company locations are required for ATCC, Epicentre and PCR enzyme (see point 6 above)

Response: We have added supplier names and locations as suggested. 

4. Figure 2-indicate what he “Rb” refers to in the legend.

Response: “Rb” refers to rabbit, we have added this to Figure 2 legend.

5. L402 i.e. (should this be italicized)?

Reponse: no

6. L416 should “in silico” be italicized?

Response: We have italicized “in silico”.

Reviewer #2

However, the amount of patients used (four) in the study is maybe a bit too small and the novelty of results limited.

Response: We agree that it would be informative to obtain this information from more patients, however, given the imperative to treat M. genitalium infection when detected in symptomatic patients it is rare to obtain consecutive specimens from individual patients, and even more uncommon to obtain corresponding serum. 

It would be nice to indicate at some point in the text the alternative nomenclature of P140 and P110 for MgpB and MgpC, respectively.

Response: We have added these alternative nomenclatures.

In the work the analysis is performed in parallel for MgpB and MgpC, except for the mapping of the variant amino acids onto the protein structure that is only done for MgpC. Since some weeks ago the structure of MgpB was published and it is now also available. In my opinion the mapping of variants should now also be done for MgpB and included in this work.

Response: The analysis of MgpB epitopes (and antibody reactivity) is planned for future experiments and will be published separately once complete. The ongoing COVID pandemic and the closure of our research building due to a radiation spill currently limit progress on this goal.

Reviewer #3

1) To identify sequence variants (L134-141 and L207-217), the authors use PCR amplification, followed by cloning and Sanger sequencing of multiple plasmids (around 75). The number of individual clones analyzed seems sufficient to obtain a representative view of the predominant sequence variants present at each time point. However, clone selection is always a matter of concern. Therefore, could Next Generation Sequencing technology have been used to analyze the different PCR amplicons? If so, do the authors think that the use of NGS could impact or improve the results presented in this study? Could the use of NGS benefit future studies addressing sequence variation of Mge during persistent infection?

Response: We appreciate this comment, and in fact attempted a similar analysis using PacBio SMRT sequencing, however, in complex mixtures of variants, PCR amplicons are predominantly heteroduplexes which was incompatible with the PacBio sequence read algorithm used at that time (See Verhey et al, PMID: 29740915). Illumina-type, short-read sequencing would not be able to distinguish between variable sequences located in the expression site from those located in the MgPars (and not expressed). Targeted Nanopore sequencing strategies are planned for future experiments. We considered that bias might be introduced during PCR and cloning and for this reason used a strategy where each variable region was amplified with two differ primer pairs for each variable region; similar sequences were obtained with both primer pairs for all four patients analyzed.

2) The Mge isolates analyzed were recovered by co-culture of processed patient urine with Vero cells (L120-133). The possible impact of co-culture with Vero cells (three or more weeks) on sequence variation is also matter of concern (L272-282). In this regard, could this step introduce some bias in the sequence variants identified? For example, selecting for adhesin variants with increased adherence capacity or some other feature important for in vitro survival. Therefore, was this co-culture strictly necessary? Can DNA from Mge be extracted/sequenced directly from a patient sample (for example the urine pellets after centrifugation)? In addition, why was only culture supernatants used to isolate DNA (L131)? I do not have experience isolating Mge clinical strains but, shouldn’t the bacteria be attached to Vero cells?

Response: As mentioned above (see Reviewer 1, comment 10), one of our future goals is to determine if antibodies (from patients or immunized rabbits) targeting a particular sequence will kill M. genitalium patient isolates expressing that sequence but not cells expressing a variant sequence. Direct analysis of patient urine specimens would include DNA from both live and dead M. genitalium which could confuse the interpretation of these experiments.

The reviewer is correct that M. genitalium cells are adherent to Vero cells, however, after 21 days of coculture the Vero cells have detached from the plastic so M. genitalium adhered to Vero cells, and M. genitalium present in the supernatant would both be included in sequence analysis, this important detail has been added to Methods.

3) The current study seems appropriate to determine the extent of sequence variation in the adhesin genes over time in men with NGU persistently infected with Mge (L174). However, the relationship between the observed changes and antigenic variation, and the impact on antibody recognition in less apparent. In this sense, all evidences are indirect. Even if amino acid changes localize (in general) to regions predicted as epitopes, no direct evidence is provided that these changes effectively avoid antibody recognition. Therefore, all conclusions on this regard must be limited and presented as a possibility rather than a fact.

To establish a solid relationship between sequence variation and antibody recognition, direct evidence is required. For example, for each particular Mge isolate, is it possible to test if patient serum from Visit 1 is less or non-reactive against MgpB or MgpC variants predominantly expressed at Visit 3? To test this, can the Mge strains expressing the adhesin variants predominant at Visit 3 be isolated and used as template (whole cell lysates) for western blotting analysis using sera from Visit 1? Alternatively, is it possible to generate Mge mutants expressing these new variants that could be analyzed by western blotting instead of G37 whole cell lysates? Moreover, could the predominant variants (MgpB/MgpC) identified at Visit 3 be expressed as recombinant proteins (heterologous expression) and used to demonstrate loss of recognition by serum from Visit 1?

Response: We agree with the reviewer’s comments and are working towards this evidence (see lines 478-481). The dependence of clinical isolates on Vero coculture for growth (and the protein contributed by these cells), and the low yield of M. genitalium from these cultures, makes it difficult to obtain enough whole cell lysate to analyze by Western blot. The experiments to express recombinant proteins of variable sequences are underway, however, these will necessarily measure reactivity to primarily linear (rather than conformational) epitopes. For this reason, in future experiments, we will replace the M. genitalium type strain (G37) variable regions with Visit 1 or Visit 3 specific sequences and measure biologic activity of patient antibodies to these engineered strains. 

We agree that our analysis provides indirect evidence that antibodies select against particular sequences; it was not our intention to present this as fact. See for example, line 36, “these findings support the hypothesis”, line 105 “consistent with a model”, and line 529-31 “Further studies are needed to determine if epitopes predicted in silico are indeed targeted by host antibodies...”. Further, we have changed “substantiate” to “support” in the Abstract (line 36).

Minor points:

Why are Figure legends embedded within the manuscript?

Response: PLOS One requires them to be there.

L64 “is affected by two mechanisms”: rather than two mechanisms, I think antigenic and phase variation are two processes or even purposes.

Response: This sentence has been revised, see response to Reviewer 1 above

L99 “Among the four”. New paragraph.

Response: Done

L135 “three (or ten weeks)”. When ten?

Response: Ten weeks refers to three serial passages of 3-4 weeks each. This has been clarified in the resubmission, see response to Reviewer 1, comment 7.

L207-217. This is Methods rather than Results?

Response: We prefer to keep this section in Results rather than Methods because we present results of sequencing two different PCR products for each region.

L224 “consistent with immune selection against specific epitopes”. Could sequence variation have other purposes? Perhaps not all sequence changes are aimed to avoid immune system recognition and are negatively/positively selected by other forces with different purposes.

Response: It has previously been suggested that variation may affect MgpBC function, for example, adherence to different host receptors. However, the recent discovery that the sialic acid binding site is composed of absolutely conserved amino acids in MgpC, and the fact that antigenic variants analyzed in vitro maintain their adherent phenotype sows doubt on that theory. Furthermore, the tremendous diversity of variable regions within and between strains (changing a few to dozens of amino acids) makes it unlikely that variants would perform different functions from each other. Nonetheless, variation may indeed have an as yet unidentified purpose other than immune evasion which is why we are careful to say that our observations are consistent with immune selection rather than proving immune selection.

L226 “the predominant sequence”. Are “predominant sequences” much more abundant than other sequences in each sample? Can the authors show the % of identification of the predominant sequence for each isolate and time point? Were mixtures often identified (other than for patient 43 Visit 1)? 

Response: This relative proportions of the different variants is displayed in Figures 3 and 4; each line represents a single sequence with the most abundant sequence indicated by the lines without colored bars. We considered presenting the various percentages of each variant, however, simple percentages don’t reflect the true similarity/diversity of the variants. For example, sequences with single amino acid change or multiple amino acid changes would both be counted as “variants” but multiple amino acid changes would likely have a larger effect on antigenicity and therefore antibody reactivity. Mixtures of variants were identified in all four patients as shown in Figures 3 and 4.

---

## [Editor Report · Decision Letter 1]

30 Sep 2020

Sequence variation and immunogenicity of the Mycoplasma genitalium MgpB and MgpC adherence proteins during persistent infection of men with non-gonococcal urethritis

PONE-D-20-17903R1

Dear Dr. Wood,

We’re pleased to inform you that your manuscript has been judged scientifically suitable for publication and will be formally accepted for publication once it meets all outstanding technical requirements.

Kind regards,

Catherine A. Brissette, Ph.D.

Academic Editor

PLOS ONE
---

## [Editor Report · Acceptance letter]

2 Oct 2020

PONE-D-20-17903R1 

Sequence variation and immunogenicity of the *Mycoplasma genitalium* MgpB and MgpC adherence proteins during persistent infection of men with non-gonococcal urethritis 

Dear Dr. Wood:

I'm pleased to inform you that your manuscript has been deemed suitable for publication in PLOS ONE. Congratulations! Your manuscript is now with our production department. 

Kind regards, 

on behalf of

Dr. Catherine A. Brissette 

Academic Editor

PLOS ONE